# A QCT View of the Interplay between Hydrogen Bonds and Aromaticity in Small CHON Derivatives

**DOI:** 10.3390/molecules27186039

**Published:** 2022-09-16

**Authors:** Miguel Gallegos, Daniel Barrena-Espés, José Manuel Guevara-Vela, Tomás Rocha-Rinza, Ángel Martín Pendás

**Affiliations:** 1Department of Analytical and Physical Chemistry, University of Oviedo, 33006 Oviedo, Spain; 2Institute of Chemistry, National Autonomous University of Mexico, Circuito Exterior, Ciudad Universitaria, Delegación Coyoacán, Mexico City C.P. 04510, Mexico

**Keywords:** hydrogen bond, interacting quantum atoms, aromaticity, QTAIM

## Abstract

The somewhat elusive concept of aromaticity plays an undeniable role in the chemical narrative, often being considered the principal cause of the unusual properties and stability exhibited by certain π skeletons. More recently, the concept of aromaticity has also been utilised to explain the modulation of the strength of non-covalent interactions (NCIs), such as hydrogen bonding (HB), paving the way towards the in silico prediction and design of tailor-made interacting systems. In this work, we try to shed light on this area by exploiting real space techniques, such as the Quantum Theory of Atoms in Molecules (QTAIM), the Interacting Quantum Atoms (IQA) approaches along with the electron delocalisation indicators Aromatic Fluctuation (FLU) and Multicenter (MCI) indices. The QTAIM and IQA methods have been proven capable of providing an unbiased and rigorous picture of NCIs in a wide variety of scenarios, whereas the FLU and MCI descriptors have been successfully exploited in the study of diverse aromatic and antiaromatic systems. We used a collection of simple archetypal examples of aromatic, non-aromatic and antiaromatic moieties within organic molecules to examine the changes in π delocalisation and aromaticity induced by the Aromaticity and Antiaromaticity Modulated Hydrogen Bonds (AMHB). We observed fundamental differences in the behaviour of systems containing the HB acceptor within and outside the ring, e.g., a destabilisation of the rings in the former as opposed to a stabilisation of the latter upon the formation of the corresponding molecular clusters. The results of this work provide a physically sound basis to rationalise the strengthening and weakening of AMHBs with respect to suitable non-cyclic non-aromatic references. We also found significant differences in the chemical bonding scenarios of aromatic and antiaromatic systems in the formation of AMHB. Altogether, our investigation provide novel, valuable insights about the complex mutual influence between hydrogen bonds and π systems.

## 1. Introduction

The hydrogen bond (HB) is one of the the most important non-covalent interactions (NCI) in nature. Since its first appearance in the chemistry parlance, back in the second decade of the twentieth century [1], HB interactions have been recognised as key factors determining the properties and structure of a wide variety of molecules and materials. Indeed, the role of HBs is known to affect countless systems, from simple molecular liquids and solids, such as water or hydrogen fluoride, to complex and intricate biomolecules. Furthermore, in recent years, a renewed interest for HB interactions has arisen within the scientific community owing to the importance of these contacts in emerging technologies such as (i) CO_2_ capture [2,3,4,5], (ii) rechargeable aqueous zinc [6,7] and aprotic Li-O_2_ batteries [8], (iii) photovoltaic cells [9,10], (iv) asymmetric catalysis [11], or (v) hydrogen production [12], among others.

As it often happens in the context of inter-molecular bonding scenarios, the complex interplay between different kinds of interactions drives the global properties of supramolecular systems. Therefore, the combination of HB with other similar or drastically different NCIs is of particular importance. We can consider, for instance, the prototypical example of water clusters. The existence of single HB donors and acceptors in H${}_{2}$O clusters has been associated with the mutual strengthening (cooperativity) of HBs (Figure 1a), whereas the occurrence of double HB donors and acceptors has been related with the reciprocal weakening (anticooperativity) of HBs (Figure 1b) [13,14,15,16,17]. Additionally, there are other instances of non-additive effects of hydrogen bonding reported in the literature, e.g., charge assisted HBs [18,19] and ion-dipole contacts [20].

As a general result, the above mentioned cooperative and anticooperative effects are the result of subtle electron fluctuations that accompany the formation of non-covalently bonded systems [21]. Some of these electron redistributions take place through σ bonds and, it is thus common to refer to them as σ-cooperative or σ-anticooperative HB effects. However, such charge transfers might also occur throughout π systems particularly those found in conjugated moieties [14,22,23,24,25,26,27]. Well-known examples of the interplay between H-bonds and conjugated π systems are Resonance-Assisted Hydrogen Bonds (RAHB) as originally proposed by Gilli et al. [28,29]. RAHBs are understood usually as the result of π-cooperative effects, which considerably strengthen HBs coupled with π bonds. On the other hand, conjugated systems and hydrogen bonds can also reveal anticooperative effects as those found, for instance, in the bicyclic fused rings of malondialdehyde [23,30] or in Resonance-Inhibited Hydrogen Bonds (RIHB) [25,26,27].

Another particularly relevant interplay between H-bonds and π systems can be found in the case of the more recently proposed Aromaticity and Antiaromaticity Modulated Hydrogen Bonds (AMHB) [31,32]. The concept of AMHB was first introduced to rationalise the apparent strengthening or weakening of HB interactions modulated by changes in aromaticity and antiaromaticity in the involved systems. Although clearly intuitive and useful, the ideas of aromaticity and antiaromaticity are built upon elusive and ill-defined chemical concepts, which hinder a quantitative and rigorous analysis. Fortunately, state-of-the-art wave function analysis methods have proved very useful in the study of electron delocalisation, which is a critical aspect in the study of aromaticity and antiaromaticity. In particular, and in the context of Quantum Chemical Topology (QCT), the Quantum Theory of Atoms in Molecules (QTAIM) [33] and the Interacting Quantum Atoms (IQA) [34] methods have been successfully exploited to investigate the mutual influence of HBs and π systems [22,23,24,25].

In this work, we make usage of the QTAIM and IQA approaches as well as electronic delocalisation indices developed within the conceptual framework of QCT to provide a detailed real-space-based picture of AMHB. For this purpose, we compared the energetics and studied the chemical bonding scenario using QCT in the formation of different AMHB molecular clusters shown in Figure 2. We emphasize the effects of the formation of different molecular clusters on pairwise inter-atomic interactions. For the sake of convenience, and considering the large computational cost of some QCT analyses, derivatives of the simple, but representative, azete and pyridinde molecules will be used as model systems in this work. It should be noticed that these molecules have already been successfully employed in the literature [31,32] as minimal models to study hydrogen bond driven dimerisation phenomena. The manuscript is organised as follows. First, we provide a brief background of the QTAIM and IQA approaches. Then, we discuss the electronic and energetic changes accompanying the dimerisation of a collection of organic scaffolds. Later, we consider the interplay between the above mentioned changes and the aromatic character of the monomers. Lastly, we examine somewhat atypical systems to finally gather the main conclusions of this work.

## 2. Theoretical Framework

### 2.1. Real Space Wavefunction Analyses

The QTAIM theory, as originally formulated by Bader [33], is a method of wave function analysis based on the topology of the electron density ρ(r), in which the real space is fragmented in a collection of attraction basins (Ω) induced by the topology of ρ(r). In QTAIM, traditional chemical ideas, such as the concept of chemical groups or fragments, atomic charges or bond orders, emerge naturally without the need of any reference. Moreover, the QTAIM partition can be performed starting either from theoretical (electronic structure calculations) or experimental (high-resolution X-ray diffraction data [35]) determinations of the electron density of the system. This combination of robustness and practicality has made QTAIM to be widely employed to shed light into a large variety of phenomena including catalysis [36,37,38], electrical conductivity [39,40,41] and aromaticity [42,43,44], to name a few.

Based on a 3D partition as that defined by QTAIM, the IQA methodology [34] divides a fully interacting non-separable quantum mechanical system into chemically meaningful interacting entities. The total electronic energies in IQA can be written as a sum of one-body (intra-atomic) and two-body (inter-atomic) terms [34,45], as:(1)$$\begin{array}{c} E=\sum_{A}E_{\mathrm{self}}^{A}+\sum_{A>B}E_{\mathrm{int}}^{AB}, \end{array}$$
where $E_{\mathrm{self}}^{A}$ is the energy of atom *A*, which includes the electron–nucleus attraction, the inter-electronic repulsion, and the kinetic energy within atom *A*. Additionally, $E_{\mathrm{int}}^{AB}$ is the total interaction energy between atoms *A* and *B*. This term encompasses all the available interaction terms between the nucleus and electrons within atoms *A* and *B*. The constituting terms of the total inter-atomic interaction between two atoms, $E_{\mathrm{int}}^{AB}$, can be regrouped to express the latter as a sum of purely covalent (i.e., exchange-correlation, $V_{\mathrm{xc}}^{AB}$) and ionic (i.e., classical, $V_{\mathrm{cl}}^{AB}$) components:(2)$$\begin{array}{c} E_{\mathrm{int}}^{AB}=V_{\mathrm{cl}}^{AB}+V_{\mathrm{xc}}^{AB}. \end{array}$$
Indeed, the IQA energy decomposition provides a particularly convenient way to study and characterise the chemical nature of the interaction among atoms in an electronic system.

### 2.2. Aromaticity

Aromaticity is a multi-factorial concept which is thought to modify and even to determine the structural, energetic, electronic, and magnetic properties of some molecules. Due to lack of an inherent Dirac observable defining it, aromaticity is usually described in terms of its effects on conjugated systems, such as enhanced thermodynamic stability or structural rigidity. Although the idea of aromaticity was conceived solely upon the interpretation of experimental results, the birth of quantum and computational chemistry motivated the development of multiple tools and techniques aiming at its quantitative analysis. One of the most common approaches to study and measure aromaticity is the nucleus-independent chemical shifts (NICS) [46,47] method, as originally proposed in 1996 by Schleyer and coworkers [46]. The NICS approach has been used for several decades to study numerous π skeletons in a variety of fields. Nevertheless, some results obtained through the NICS descriptor have turned to be highly questionable [48,49,50], even contradicting, in some cases, other aromaticity measures based on reactivity [51]. Among many other criteria exploited to quantify aromaticity we can find (i) structural indices, which evaluate bond equalisation [52], (ii) energy decomposition analyses which require a reference molecule [53], and (iii) electronic descriptors which evaluate the amount of electron delocalisation among the atoms forming a cyclic structure. The last-mentioned set includes a number of methods that have been developed relying on the partition of the electronic density offered by QTAIM, such as the Para Delocalisation Index (PDI) [54], the Aromatic Fluctuation Index (FLU) [55], or the Multicenter Index (MCI) [56]. The PDI and FLU approaches provide an estimate of the aromaticity of a system in terms of the electron delocalisation within the cyclic skeleton, whereas the MCI method arises from a generalised population analysis leading to a many-atoms bond index. In the present work, we have made use of some electronic delocalisation-based descriptors, such as the MCI or the FLU indices, to quantify the changes in aromaticity and antiaromaticity of each molecule upon the formation of the corresponding dimer. We have chosen the FLU and MCI indicators to account for the changes in aromaticity and antiaromaticity upon interaction of the monomers under consideration given their proven accuracy and reliability [57]. Furthermore, these methods are fully compatible with the rest of the QCT analyses performed in this report. Unfortunately, the PDI method is not applicable to some of the examined systems herein, because it can only be used for six-membered rings. Finally, we also used the IQA partitioning to study in detail the energetic changes accompanying the generation of the molecular clusters shown in Figure 2, with a particular emphasis on their role in HB formation.

## 3. Computational Details

The structures of the hydrogen-bonded dimers in Figure 2 were optimised in the gas phase and the resultant approximate wavefunctions and electron densities were afterwards dumped for further analysis. All geometry optimisations were performed using the Orca quantum chemistry package version 5.0.3 [58] using the PBE0 hybrid functional [59] along with the Def2-TZVP basis set [60] and the atom-pairwise dispersion correction with the Becke–Johnson damping scheme [61,62]. For the sake of computational efficiency, the Resolution of Identity (RI) approximation was used for the Coulomb integrals with the default COSX grid for HF exchange, as implemented in Orca [58]. On the other hand, the auxiliary Def2 basis set was used for the RI-J approximation. The combination of such an exchange-correlation functional and basis set has proven [32] suitable for the characterisation of the systems under study. Moreover, DNLPO-CCSD(T)/def2-QZVPP single point calculations were performed on the optimised geometries of the monomers and dimers in order to test the accuracy of our DFT results. An extrapolation to the complete basis set limit was performed through the def2-SVP/def2-TZVP scheme as implemented in Orca [58] so as to ameliorate the Basis Set Superposition Error (BSSE). The nature of the stationary points (corresponding to local minima of the potential energy surface) was characterised through the computation of the corresponding harmonic frequencies. QTAIM and IQA calculations were performed using the AimAll [63] and Promolden codes [64]. The exchange-correlation energy was partitioned as indicated in reference [65]. Finally, all aromaticity indices discussed along this manuscript were computed using the ESI-3d code [66]. We have denoted the dimers with (i) the hydrogen bond Acceptor Contained within the Ring and (ii) the hydrogen bond Donor Contained within the Ring, as ACR and DCR, respectively. For the ACR azet-2-1H-one (AZH) dimer, we performed a geometry-constrained optimisation to ensure the attainment of the right tautomer of the constituting monomers. On the other hand we observed for the DCR-AZH dimer a structure considerably deviated from planarity as opposed to the rest of the systems. Additional information such as optimised structures, electronic energies, and a more complete survey of IQA and QTAIM (e.g., IQA energy of different groups as well as other QTAIM descriptors) can be found in the electronic Appendix A.

## 4. Results and Discussion

### 4.1. General Energetic Changes Induced by HB Formation

We consider first the differences in dimerisation energies of the different systems shown in Figure 2. Table 1 reports the values of ΔΔE,
(3)$$\Delta \Delta E\left(Y_{2}\right)=\Delta E\left(Y_{2}\right)-\Delta E\left(R_{2}\right),$$
in which $\Delta E(Y_{2})$ is the energy change associated to the process
(4)$$Y+Y⇌Y\cdots Y,\;\Delta E(Y_{2}),$$
and R is the corresponding reference system used for ACR and DCR, namely the dimers of formamide (NCO) and formamidine (NCN) in the corresponding ACR and DCR form. A negative/positive value of $\Delta \Delta E(X_{2})$ indicates a stronger/weaker interaction in X⋯X with respect to the reference complex R⋯R.

The straightforward comparison of the DFT and CC values for ΔΔE reveals that both levels of theory are in good agreement concerning the sign and magnitude of ΔΔE. These observations indicate that our DFT results offer a reliable picture of the energetics of the binding phenomena under study. As the footnote of Table 1 reports, all the values for ΔE are negative, pointing, as expected, to stabilising dimerisation contacts in all the investigated dimers. Furthermore, the easiness of the complexation seems to be driven, as expected, by the hydrogen bond formation as reflected by the correlation of the binding energies with the ρ at the bond critical point of the HB contacts (see Appendix A). We also note that the N−C=N bonding pattern leads to lower binding energies, due to the larger acidity of H atoms bonded to oxygen. The AMHB [32] interpretation of the sign of ΔΔE in Table 1 states that the ACR dimers AZH and AZA display either an increase in aromaticity or a decrease in antiaromaticity, respectively, as a consequence of the formation of the investigated H-bonds. Ditto for the DCR clusters 2HP and 2AP. On the other hand, the ACR complexes 2HP and AZA along with the DCR systems AZH and AZA exhibit the opposite behaviour. We consider now QCT analyses to further dissect these energetic trends.

### 4.2. Quantum Chemical Topology Analyses

In order to further deepen into the origin of the observed trends in the evolution of the binding energies reported in Table 1, we examined the non-covalent interactions established between both monomers using QCT techniques. For the sake of convenience, the nomenclature shown in Figure 3 will be used to refer to the atoms involved in the inter-molecular bonding pattern of these dimers. We first consider the electron redistribution of electron charge due to the formation of the investigated H-bonds. We point out that the formation of an HB is associated with a reorganisation of the electronic density of the moieties involved in this interaction. There is, indeed, a transfer of electron charge from the HB acceptor to the HB donor, with the proton acting as a bridge. For small HB dimers such as (H_2_O)_2_ or (HF)_2_, two of the simplest HBs, such charge displacement makes the HB acceptor a better proton donor. Ditto for the HB donor becoming a better proton acceptor. Notwithstanding, the present work deals with dimers where each molecule acts simultaneously as an HB donor and an HB acceptor and hence, there is no effective charge transfer between the monomers. However, the presence of an HB induces a rearrangement of the electron density that interacts with the π clouds in each molecule of the studied systems.

Let us start by examining the major changes undergone upon the dimerisation of the non-aromatic reference systems: formamide and formamidine. The complexation process is accompanied by a significant electron redistribution, as reflected by the change in the QTAIM atomic charges, collected in Table 2. The formation of the non-covalent interactions leads, in both cases, to a noticeable electron enrichment of the D and A atoms (between 0.03 and 0.09 electrons) at the cost of decreasing the electron population of the H atom by ≈0.04–0.09 electrons. On the other hand, the central C atom undergoes a noticeable change in its average electron number of −0.09–0.01 a.u. depending on the nature of the acceptor moiety.

These observations, and with the particular exception of the bridging C atom, are very similar for both NCO and NCN bonding patterns and evidence a conspicuous rise in the polarisation of the system due to the formation of the corresponding dimers. Such an increase in the local polarisation of the terminal atoms enhances the electrostatic interaction in the HB contacts, as reflected by the large classical components of the A···H interaction as reported in Appendix A.

We also considered the change in the number of electrons shared among bonded atoms, as measured by the delocalisation index (DI) (see Appendix A for more details), as gathered in Table 3. The D–H bond order decreases significantly (≈0.16–0.25) upon dimerisation, thus weakening the covalent component to the D–H interaction, as evidenced by the prominent destabilisation of ≈30–40 kcal/mol found for $V_{\mathrm{xc}}^{D–H}$. A similar, yet more subtle, weakening of the covalent component can also be observed for the C–A bond. On the other hand, the DI(D–C) is increased by 0.07–0.10 electron pairs, going from a single D–C bond to a slightly higher bond order (≈1.1 in the general case). These results point out that hydrogen bonding reinforces the D–C double bond character at the expense of decreasing that of the C–A interaction. We observed a similar effect in our analysis of RAHB in which the DI corresponding to double bonds decrease while that of single bonds have the opposite behaviour after the formation of the RAHB [22].

This last observation is fulfilled for all the systems and suggests that the formation of the dimers may trigger two opposed effects. Because the A–C bond is contained within the ring in ACR dimers, and ΔDI(C–A) < 0 as indicated in Table 3, we would expect that the formation of the H-bond would decrease the number of π electrons in the associated ACR cyclic structures as represented in Figure 4. On the contrary, the D–C bond is included in the cyclic structures of DCR dimers, and ΔDI(D–C) > 0 (Table 3), then the number of π electrons must increase in the DCR dimers due to the formation of the H-bond. Accordingly, Appendix A indicates that the group energy of the ACR/DCR rings increase/decrease upon the formation of the corresponding dimers. These changes in electron delocalisation affect the aromaticity and antiaromaticity of the investigated systems as discussed below.

### 4.3. Perturbation of the Aromaticity of the π Skeleton

We consider now the interplay between aromaticity and antiaromaticity with the inter-molecular HB contacts of Figure 2. Table 4 gathers the change in the aromaticity indices of the intra-molecular π skeleton upon dimerisation, as measured by the MCI and FLU indices (further details about these indices can be found in Appendix A).

Before discussing in detail the changes in the aromatic character of the spectator groups, it may be enlightening to provide a grasp of the FLU and MCI aromaticity indices. The former measures the electron sharing between neighbouring atoms in a ring as well as its similarities between the constituents of the cyclic structure. Thus, a FLU value of zero corresponds to an “ideal” aromatic system, while positive values evidence a deviation from aromaticity. On the other hand, the MCI index measures the collective electron delocalisation along a collection of M centres. As opposed to the FLU, large MCI values suggest a high aromatic character, whereas any other situation usually results in vanishing MCI indexes. Although these metrics were specifically designed to measure aromaticity, they have been successfully used to study antiaromaticity as well [67]. We used the Hückel rule to assign the aromatic or antiaromatic character of the examined monomers as shown in Figure 5. The aromaticity metrics of the monomers (see Appendix A) are in agreement with the aromaticity or antiaromaticity label as determined by the Hückel rule. Indeed, the DCR form of AZH and AZA is more aromatic than their ACR counterparts. Likewise, the ACR tautomer of 2HP and 2AP is more aromatic than the corresponding DCR structures.

We discuss now how the aromaticity and antiaromaticity of the corresponding monomers change due to the formation of the HB interactions. We mention that apart from the DCR-AZH scaffold, all dimers adopt a nearly fully planar disposition which is key for the delocalisation of π electrons and it is optimal for the formation of the AMHB. Except for a slight discrepancy within the results of (i) the FLU on one hand and (ii) MCI and NICS [32] on the other for the change in the aromatic character between the DCR systems AZH and AZA, there is a good agreement between the computed sign of ΔΔE and the changes of aromaticity and antiaromaticity in the examined systems, as shown in Figure 6.

In short, we observe that the condition ΔΔE<0, i.e., a more favourable formation of the HB with respect to the reference is related with a reduction in the antiaromatic character of the monomers. Correspondingly, when ΔΔE>0, i.e., a less favourable HB with respect to the reference is accompanied by a reduction of the aromaticity of the monomers. These observations based on QCT are consistent with those of Wu and coworkers [32]. We note some flexibility in the interpretation of these results. Namely, the decrease in aromaticity in the HB formation of the AZA and AZH dimers in DCR configuration was interpreted in Reference [32] as an intensification of antiaromaticity (Figure 7). Similarly, the decrease in antiaromaticity of the DCR systems 2HP and 2AP after the generation of the corresponding dimers was interpreted as an HB reinforced by an increase in aromaticity [32]. This observation suggests that aromaticity and antiaromaticity can be put in a similar scale using electron delocalisation tools within the conceptual framework of QCT.

We consider now a further QCT description of the aromatic and antiaromatic moieties considered herein (Figure 5). We observed important differences concerning the atoms directly involved in the H-bond depending on the aromatic or antiaromatic character of the interacting monomers. For the sake of clarity, we will generally refer to the changes of QTAIM and IQA properties upon dimerisation, with respect to the NCN or NCO reference systems.

As expected, the relative change in the atomic charges and delocalisation indices reported in Table 2 and Table 3 have a notable impact on the covalent and ionic components to the total IQA interaction energies in the atoms directly involved in the H-bond. Figure 8 collects the change in the classical and exchange-correlation components of $E_{\mathrm{int}}$ for the atoms entailed directly in the inter-molecular contact, reported relative to the NCO or NCN reference systems. The aromatic scaffolds (see Figure 5) seem to consistently stabilise the $V_{\mathrm{xc}}^{C–A}$ and $V_{\mathrm{xc}}^{D–H}$ contributions over the rest of the interactions. The classical term of other interactions shows, on the other hand, trends that are more complicated to interpret. The interplay between these two contribution results in a net stabilisation of the C–A component at the cost of partially disrupting the D–C bond, when measured with respect to their references.

Antiaromatic systems have a different behaviour concerning the weakened and strengthened interactions in the inter-molecular region due to the formation of the examined H-bonds. These monomers (Figure 5) generally strengthen both the classical and exchange-correlation components of the H···A contact further than the reference systems. This fortifying of the HB interaction is accompanied by a noticeable destabilisation of the covalent component of the C–A bond, which, as previously discussed, is more strengthened in the aromatic compounds than it is in the reference compounds.

### 4.4. The Peculiar Case of the AZH (DCR) Dimer

As previously mentioned, all of the dimers with the exception of AZH (DCR) exhibit a planar or quasi-planar structure. However, the lowest local energy minimum found for the last-mentioned compound adopts a distorted conformation. Based on the aforementioned observations, one might conjecture at first glance that such a geometrical distortion could be understood as a way to alleviate the reduction of aromaticity induced by the formation of the dimers. To explore this idea, two additional conformational isomers were studied, as represented in Figure 9. Both the bent–trans bent–cis structures are bona fide local minima with a non-planar geometry. We performed a constraint optimisation in order to obtain the corresponding planar AZH (DCR) isomer. As shown in Table 5, the bent isomers are almost degenerate in terms of energy, being the bent–cis structure slightly more stable by ≈0.5 kcal/mol. On the other hand, the planar structure is ≈7 kcal/mol less stable than the latter.

The aromaticity metrics reported in Table 5 indicate that the restriction of the 4-membered rings in AZH to remain in a plane would lead to a further reduction of the aromaticity of the dimer. Moreover, the distortion of the low energy (bent) conformations has a dramatic impact on the QTAIM descriptors. Indeed, the changes in the atomic charges and the delocalisation indices are drastically increased, as reflected by the trends in ΔQ and ΔDI values, gathered in Table 6. As it can be seen from the changes in the delocalisation indices in the same chart, the planar isomers lead to a very prominent decrease of the D–H and C–A bond orders while promoting the delocalisation of electrons involved in the D–C and H–A interactions.

Further information can be obtained through the analysis of the IQA interaction energies, as gathered at the bottom of Table 6. The trends in the $E_{\mathrm{int}}$ energies reveal that distorting the more stable bent geometry stabilises all the pairwise interactions involved between the terminal atoms participating in the binding. This observation is particularly prominent for the D–C and H–A bonds. The interplay between the exchange-correlation and electrostatic contributions also leads to a moderate stabilisation of the D–H and C–A bonds despite the already mentioned decrease in the DI index. Thus, and in agreement with the aforementioned trends, forcing the planarity of the system further boosts the HB contacts as well as the π cloud of electrons through the promotion of the “in-ring” resonant structure. Such an effect is consistent with the enhancement of the anti-aromatic character of the system (top of Figure 7) along with the decrease of the net binding affinities, despite the more favourable hydrogen bonding established between the monomers.

## 5. Conclusions

We presented an analysis of aromaticity and antiaromaticity modulated hydrogen bonds using quantum chemical topology tools, namely the QTAIM, the IQA energy partition as well as the electronic delocalisation indicators FLU and MCI. For this purpose, we considered rings containing either the H-bond acceptor (ACR) or the H-bond donor (DCR). Our results show how the formation of the investigated H-bonds can trigger subtle electronic rearrangements with a quite significant impact in the stability and properties of the involved interacting systems. We described large changes in QTAIM charges and electron delocalisation indices along with their accompanying classical and exchange-correlation components of the IQA interaction energies related with the formation of these HB clusters. We also found fundamental differences within the ACR and DCR systems, for example, the weakening and strengthening of double bonds within the cyclic structures of ACR and DCR, a condition which leads to the destabilisation and stabilisation of the rings in these systems. Additionally, we related the enhancement and impairment of the examined H-bonds with respect to non-aromatic (i.e., non-cyclic) structures with changes in the aromatic and antiaromatic character of the system. We observe that reductions in aromaticity can be interpreted as increases in antiaromaticity and vice versa. Therefore, our results indicate that aromaticity and antiaromaticity can be considered on a common scale using QCT tools. Our results also point that the deviation from planarity of specific AMHB clusters could be related with a trend of the system to ameliorate a reduction in aromaticity. Overall, we expect the results of our investigation to provide novel useful insights about the intricate interplay among H-bond and π systems.

## Figures and Tables

**Figure 1 molecules-27-06039-f001:**
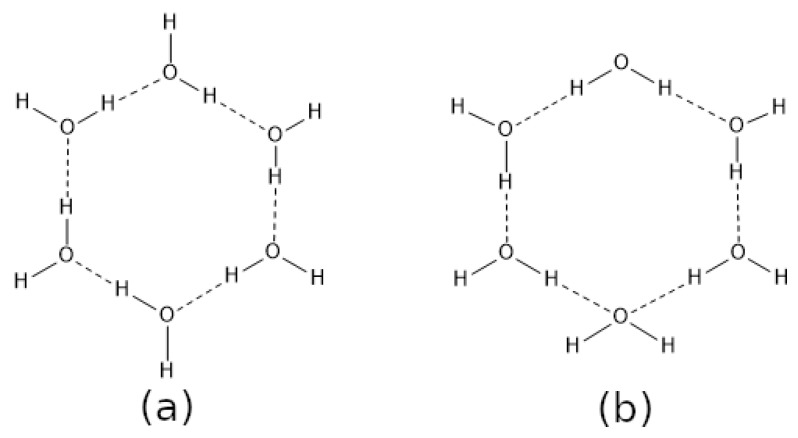
(**a**) Homodromic and (**b**) antidromic cycles within the structure of (H_2_O)_6_. These two motifs are, respectively, related to cooperative and anticooperative hydrogen bonding effects.

**Figure 2 molecules-27-06039-f002:**
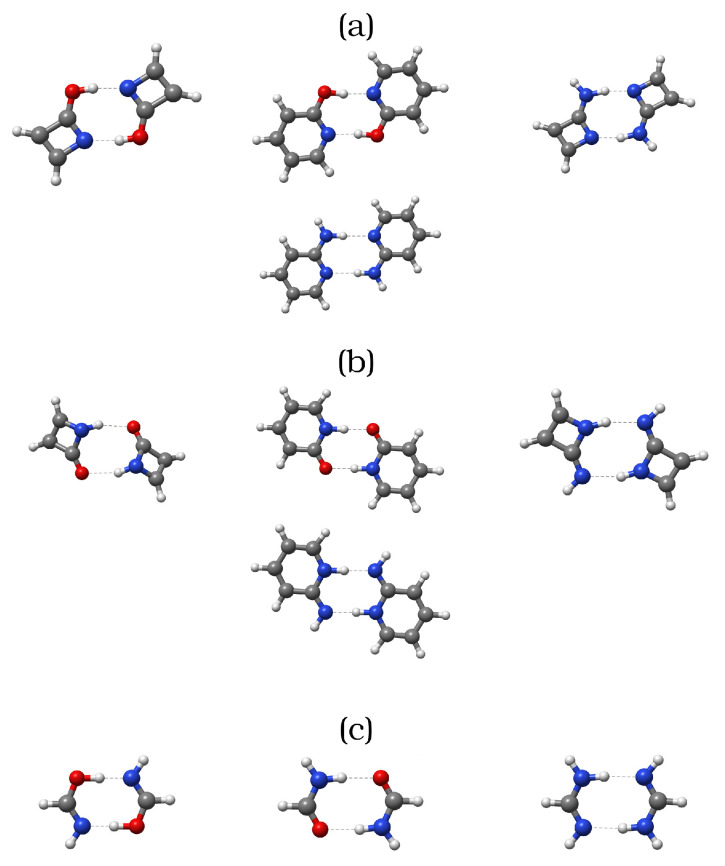
Systems examined throughout this investigation. (**a**) Dimers with the hydrogen bond acceptor contained within the ring (ACR): azet-2-ol (AZH), 2- hydroxypyridine (2HP), azet-2-amine (AZA) and 2-aminopyridine (2AP). (**b**) Dimers with the hydrogen bond donor contained within the ring (DCR): azet-2(1*H*)-one (AZH), pyridin-2(1*H*)-one (2HP), azet-2-amine (AZA) and pyridin-2(1*H*) imine (2AP). (**c**) Representation of the ACR and DCR dimers of formamide (NCO) and formamidine (NCN), used as reference. In NCN, the ACR and DCR tautomers are indistinguishable.

**Figure 3 molecules-27-06039-f003:**
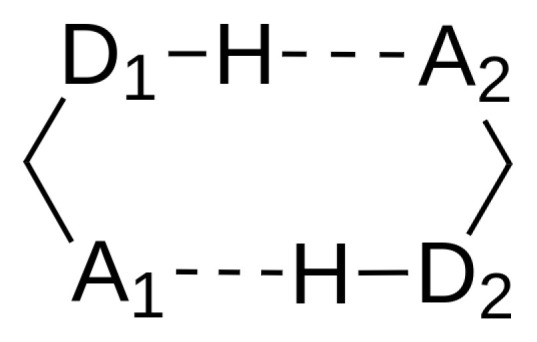
Hydrogen bond connectivity pattern involved in the formation of the dimers. A and D stand for the acceptor and donor HB moieties, respectively.

**Figure 4 molecules-27-06039-f004:**
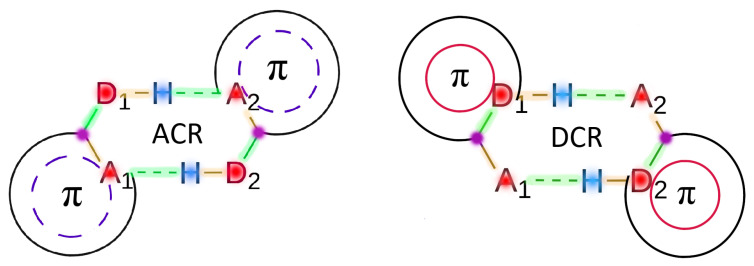
Representation of the major electronic changes induced by the dimerisation process in systems with (i) the hydrogen bond acceptor contained within the ring (ACR) and (ii) the hydrogen bond donor contained within the ring (DCR) displayed in the left and right parts of the Figure, respectively. Red and blue colors indicate QTAIM atoms with electronic charge accumulation and depletion, respectively, due to the formation of the molecular cluster. Ditto for green and orange bonds, employed to highlight an increase or decrease in the DI values. Purple is used to show not clearly established scenarios in this regard.

**Figure 5 molecules-27-06039-f005:**
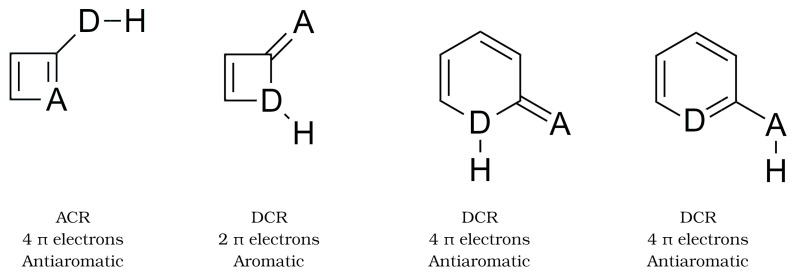
Aromatic and antiaromatic character of the monomers shown in Figure 2 determined with Hückel rule.

**Figure 6 molecules-27-06039-f006:**
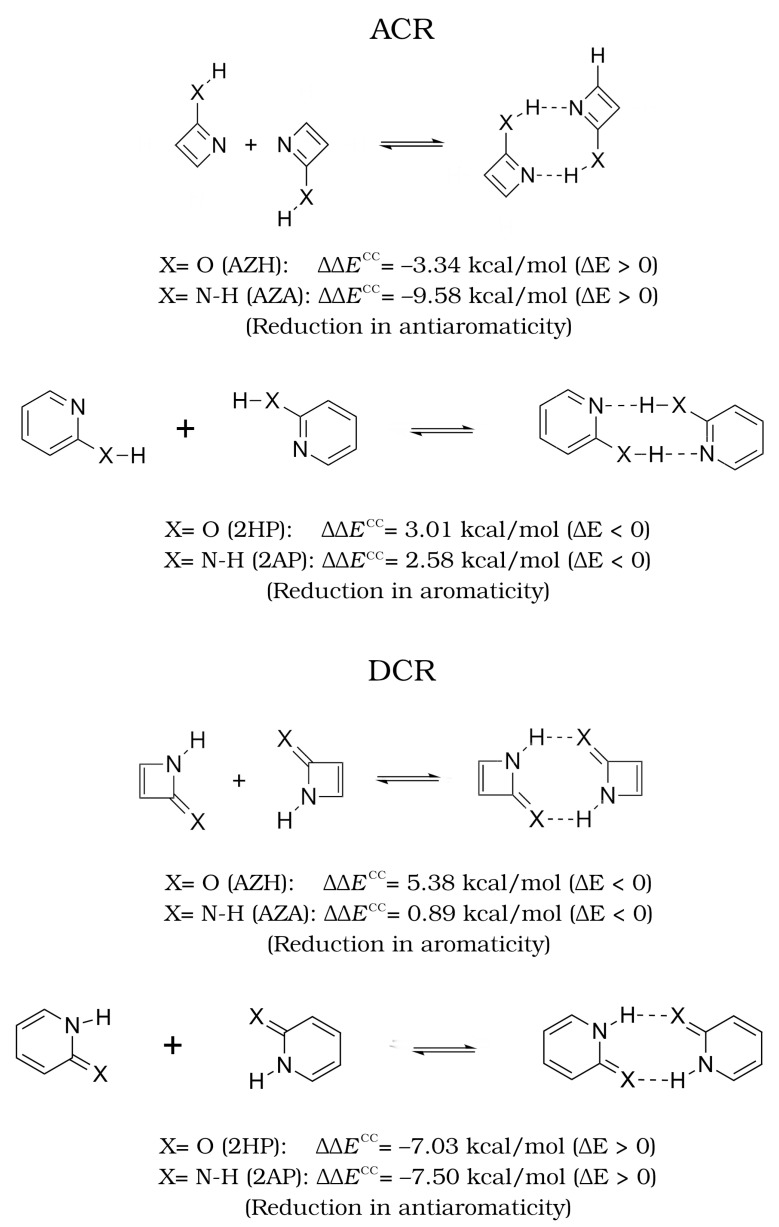
Values of ΔΔE and changes in aromaticity and antiaromaticity (ΔΓ) for the AMHB investigated herein.

**Figure 7 molecules-27-06039-f007:**
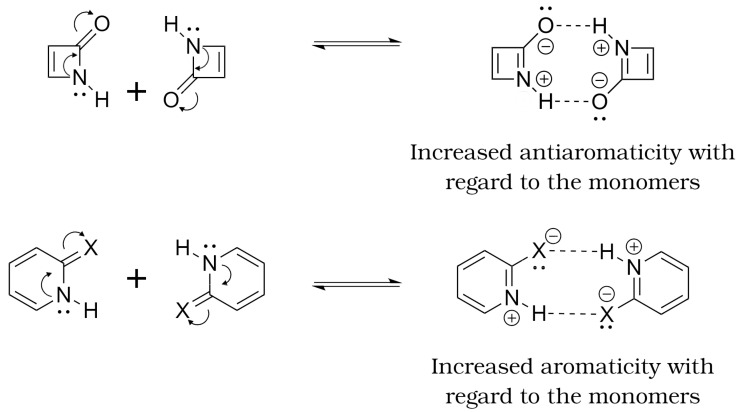
Alternative interpretation of the AMHB in the DCR dimers compared to that offered in Figure 6 as increases in aromaticity and antiaromaticity in the (i) 2HP and 2AP on one hand and (ii) the AZA and AZH on the other, respectively.

**Figure 8 molecules-27-06039-f008:**
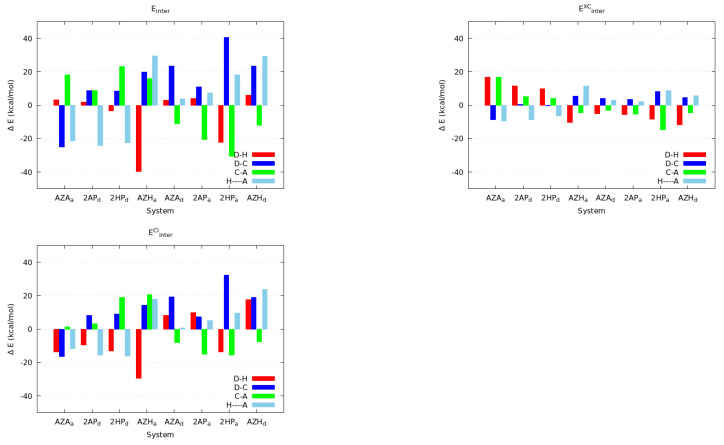
Relative changes in the IQA interaction energies upon dimerisation, all of the values are reported relative to their NCO or NCN reference systems. The tautomeric forms ACR and DCR are indicated by the a and d subscripts, respectively.

**Figure 9 molecules-27-06039-f009:**
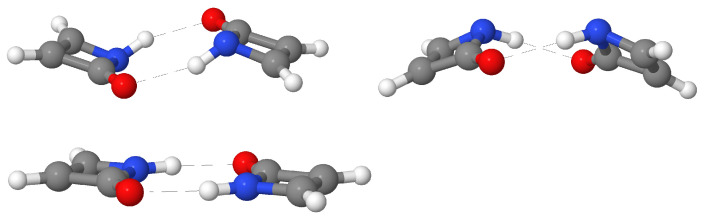
Different isomers of the AZH (DCR) dimers: bent–trans, bent–cis, planar.

**Table 1 molecules-27-06039-t001:** Values of ΔΔE, as defined in Equation (Equation 3), computed in the DFT and CC approximations described in the main text. NCO and NCN denote formamide and formamidine, respectively, the reference systems shown in Figure 2c. All values are reported in kcal/mol.

System	$\Delta \Delta E^{\mathrm{DFT}}$	$\Delta \Delta E^{\mathrm{CC}}$	System	$\Delta \Delta E^{\mathrm{DFT}}$	$\Delta \Delta E^{\mathrm{CC}}$
2HP (ACR)	5.06	3.01	2HP (DCR)	−7.11	−7.03
NCO (ACR)	0.00	0.00	NCO (DCR)	0.00	0.00
AZH (ACR)	−2.82	−3.34	AZH (DCR)	6.40	5.38
2AP (ACR)	3.22	2.58	2AP (DCR)	−8.16	−7.50
NCN (ACR)	0.00	0.00	NCN (DCR)	0.00	0.00
AZA (ACR)	−10.05	−9.58	AZA (DCR)	2.13	0.89

The corresponding values for Δ*E*/kcal·mol^−1^ for the references in Formula (4) are NCO (ACR): Δ*E*^DFT^ = −35.45, Δ*E*^CC^ = −30.10; NCN (ACR): Δ*E*^DFT^ = −16.75, Δ*E*^CC^ = −13.64; NCO (DCR): Δ*E*^DFT^ = −15.93, Δ*E*^CC^ = −13.59; NCN (DCR): Δ*E*^DFT^ = −16.75, Δ*E*^CC^ = −13.64.

**Table 2 molecules-27-06039-t002:** Change in the QTAIM electron populations of the atoms involved in the HB contacts upon the formation of the dimers for (i) the hydrogen bond Acceptor Contained within the Ring (ACR) and (ii) the hydrogen bond Donor Contained within the Ring (DCR) cases. The labelling of the atoms is shown in Figure 3. All values are reported relative to the monomers which were used as reference. Atomic units are used throughout.

ACR				
System	ΔQ(D)	ΔQ(H)	ΔQ(A)	ΔQ(C)
2HP	−0.087	+0.049	−0.067	+0.069
NCO	−0.085	+0.041	−0.072	+0.085
AZH	−0.089	+0.065	−0.077	+0.045
2AP	−0.040	+0.070	−0.021	+0.019
NCN	−0.051	+0.088	−0.024	+0.008
AZA	−0.061	+0.113	−0.063	−0.002
DCR				
System	ΔQ(D)	ΔQ(H)	ΔQ(A)	ΔQ(C)
2HP	−0.070	+0.105	−0.024	−0.040
NCO	−0.052	+0.085	−0.025	−0.007
AZH	−0.037	+0.064	−0.024	−0.011
2AP	−0.063	+0.101	−0.031	−0.012
NCN	−0.051	+0.088	−0.024	+0.008
AZA	−0.030	+0.074	−0.034	−0.007

**Table 3 molecules-27-06039-t003:** Change in the electron delocalisation index of the atoms involved in the HB contacts (Figure 3) upon the formation of the dimers with (i) the hydrogen bond acceptor contained within the ring (ACR) and (ii) the hydrogen bond donor contained within the ring (DCR). These changes are computed with respect to the values of the monomers, which were used as references. Atomic units are used throughout.

ACR				
System	ΔDI(D–H)	ΔDI(D–C)	ΔDI(C–A)	ΔDI(H–A)
2HP	−0.207	+0.056	−0.064	+0.152
NCO	−0.246	+0.100	−0.171	+0.197
AZH	−0.216	+0.079	−0.136	+0.134
2AP	−0.151	+0.051	−0.034	+0.103
NCN	−0.181	+0.073	−0.076	+0.117
AZA	−0.262	+0.131	−0.167	+0.165
DCR				
System	ΔDI(D–H)	ΔDI(D–C)	ΔDI(C–A)	ΔDI(H–A)
2HP	−0.209	+0.065	−0.091	+0.132
NCO	−0.162	+0.069	−0.070	+0.099
AZH	−0.103	+0.034	−0.043	+0.066
2AP	−0.234	+0.067	−0.106	+0.163
NCN	−0.181	+0.073	−0.076	+0.117
AZA	−0.152	+0.043	−0.053	+0.099

**Table 4 molecules-27-06039-t004:** Change in the MCI and FLU aromaticity indices, along with the change in aromatic/antiaromatic character (Γ), induced by the formation of the dimers in Figure 2. If ΔΓ>0, there is either (i) an increase of aromaticity or (ii) a reduction of antiaromaticity; vice versa when ΔΓ<0.

System	ΔMCI (a.u.)	ΔFLU (a.u.)	ΔΓ	System	ΔMCI (a.u.)	ΔFLU (a.u.)	ΔΓ
2HP (ACR)	−0.008	+0.001	−	2HP (DCR)	+0.007	−0.010	+
AZH (ACR)	+0.001	−0.017	+	AZH (DCR)	−0.001	−0.005	−
2AP (ACR)	−0.004	+0.001	−	2AP (DCR)	+0.007	−0.010	+
AZA (ACR)	+0.004	−0.021	+	AZA (DCR)	−0.001	−0.007	−

**Table 5 molecules-27-06039-t005:** Energies and aromaticity indices for the different isomers of the AZH (DCR) dimers. All values are reported relative to the most stable isomer (bent–cis).

Isomer	ΔE (kcal/mol)	ΔFLU (a.u.)	ΔMCI (a.u.)
bent–trans	+0.48	+0.0006	−0.0001
bent–cis	0.00	0.0000	0.0000
planar	+7.38	+0.0094	−0.0038

**Table 6 molecules-27-06039-t006:** Change in the QTAIM charges and electron delocalisation indices, in atomic units, upon the formation of the different AZH (DCR) dimers. Some selected changes of IQA interaction energies (in kcal/mol) are reported as well.

QTAIM charges				
System	ΔQ(D)	ΔQ(H)	ΔQ(A)	ΔQ(C)
bent–trans	−0.039	+0.063	−0.024	−0.007
bent–cis	−0.037	+0.064	−0.024	−0.011
planar	−0.185	+0.127	−0.030	+0.065
Delocalisation indices				
System	ΔDI(D–H)	ΔDI(D–C)	ΔDI(C–A)	ΔDI(H–A)
bent–trans	−0.099	+0.035	−0.043	+0.064
bent–cis	−0.103	+0.034	−0.043	+0.066
planar	−0.169	+0.070	−0.068	+0.074
IQA energy partition				
System	$\Delta E_{\mathrm{int}}^{D–H}$	$\Delta E_{\mathrm{int}}^{D–C}$	$\Delta E_{\mathrm{int}}^{C–A}$	$E_{\mathrm{int}}^{H–A}$
bent–trans	−12.80	−21.21	+11.99	−87.41
bent–cis	−12.87	−18.43	+13.47	−88.66
planar	−40.98	−121.17	−1.07	−106.99

## Data Availability

Structures of the studied systems are reported in the Electronic Appendix A.

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
