# Peer review of "A QCT View of the Interplay between Hydrogen Bonds and Aromaticity in Small CHON Derivatives"

_molecules, 2022, doi:10.3390/molecules27186039_

Round 1

Reviewer 1 Report

Present paper is devoted to theoretical investigation of Aromaticity and Antiaromaticity Modulated Hydrogen Bonds phenomena. For examination of given phenomena several well established techniques (QTAIM, IQA, FLU, MCI) were applied. 

Title of manuscript seems too general. I am on the same side with authors, and strongly believe that a wide range of molecules possess the same phenomena. However in present manuscript only derivatives of azete and pyridine were considered. Thus, it seems reasonable to avoid overgeneralization and highlight the concrete derivatives under consideration in the main title.

The addition of motivation for the choice of azete and pyridine derivatives should make the present study clearer for readers. 

Authors extensively applied QTAIM analysis. It is a reliable and robust method. As I understand authors focused mainly on delocalisation index results. However given method allows one to investigate a lot of other useful descriptors ( electron density, total energy at bond critical points, elliplicity etc). Potentially investigation of correlation between several qtaim descriptors may be a good addition to work.

Performing EDA (Energy Decomposition Analysis) can be a fruitful expanding of present investigation. However it is very time-consuming calculation, thus, it is up to authors to perform such calculation or not. 

Please add the version of Orca to computational parts.

Finally, it is nice work which is suitable for publication after performing relevant improvements. 

Reviewer 2 Report

The article “A QCT view of the interplay between hydrogen bonds and aromaticity” by Gallegos et al. describes using various real-space electron density-based approaches effects of hydrogen bond formation on (anti)aromaticity of interacting partners.

The manuscript is very well written. The research is convincingly conducted as well as the conclusions drawn by the authors. Indeed, a collection of descriptors was used to characterize the effect of aromaticity on hydrogen bonds energetics, as well as effect of HB formation on aromaticity, all accounting for the cases when HB donors or acceptors are contained in the ring. Reference hydrogen bonded dimers were considered, reproducing interactions found between studied aromatic compounds, but without contributions of the rings. This is of course methodologically sound.

I have only minor remarks / suggestions for the authors:

The Delocalization Index (DI) is used throughout the manuscript, but unlike FLU or MCI it is poorly introduced. Instead, the Para Delocalization Index (PDI) is mentioned in the introduction just to say it is not applicable here. I think DI should be briefly introduced in Supplementary Material, as it is the case for other descriptors. This work discussing HB/aromaticity might target a much broader audience than the QTAIM/QCT/Quantum Crystallography communities, hence the importance of clearly describing all quantum-based approaches used here.   

Not a big deal, but DFT and CCSD results are given, apparently to validate DFT calculations. CC results are not interpreted per se. Why not simply adding the CC values in SI, reducing this way Table 1 which contains a pivotal result of the study?

The sentence “We observed a similar effect in our analysis of RAHB in which the DI corresponding to double bonds decrease while that of single bonds have the opposite behaviour after the formation of the RAHB” needs to be supported by a reference.

The way G and DG are obtained should be explained.

Figure 6 has two problems. (1) Molecules are drawn with -X-H groups and X is said to be either O (ok) or N-H (not ok). (2) “DH” can be seen in the picture… Initially I was puzzled wondering where were these enthalpy variations (and their strange signs) come from .. to eventually realize they are actually DG! Please fix this figure 6.

Many interesting data can be found in SI but are not even mentioned in the main text. I refer in particular to the spectacular Figure S3, suggesting a linear relationship between dimers binding energy and ED on (which?) HB critical point(s).

Table 2 and table S10 are duplicates. 

Page4, lign 89 : a typo with “from”

Page6, lign 191 : pretty sure ‘y’ means ‘and’ in Spanish

Round 2

Reviewer 1 Report

Authors have made all appropriate improvements. Thus, work is suitable for publication.